# Learning Dynamic Belief Graphs to Generalize on Text-Based Games

**Ashutosh Adhikari**[†][*]   **Xingdi Yuan**[♡][*]   **Marc-Alexandre Côté**[♡][*]
**Mikuláš Zelinka**[‡]   **Marc-Antoine Rondeau**[♡]
**Romain Laroche**[♡]   **Pascal Poupart**[†][¶]   **Jian Tang**[♠][♣]
**Adam Trischler**[♡]   **William L. Hamilton**[◇][♣]

[†]University of Waterloo   [♡]Microsoft Research, Montréal   [‡]Charles University
[♣]Mila   [◇]McGill University   [♠]HEC Montréal   [¶]Vector Institute
eric.yuan@microsoft.com

## Abstract

Playing text-based games requires skills in processing natural language and sequential decision making. Achieving human-level performance on text-based games remains an open challenge, and prior research has largely relied on hand-crafted structured representations and heuristics. In this work, we investigate how an agent can plan and generalize in text-based games using graph-structured representations learned end-to-end from raw text. We propose a novel graph-aided transformer agent (GATA) that infers and updates latent *belief graphs* during planning to enable effective action selection by capturing the underlying game dynamics. GATA is trained using a combination of reinforcement and self-supervised learning. Our work demonstrates that the learned graph-based representations help agents converge to better policies than their text-only counterparts and facilitate effective generalization across game configurations. Experiments on 500+ unique games from the TextWorld suite show that our best agent outperforms text-based baselines by an average of 24.2%.

## 1  Introduction

Text-based games are complex, interactive simulations in which the game state is described with text and players act using simple text commands (e.g., `light torch with match`). They serve as a proxy for studying how agents can exploit language to comprehend and interact with the environment. Text-based games are a useful challenge in the pursuit of intelligent agents that communicate with humans (e.g., in customer service systems).

Solving text-based games requires a combination of reinforcement learning (RL) and natural language processing (NLP) techniques. However, inherent challenges like partial observability, long-term dependencies, sparse rewards, and combinatorial action spaces make these games very difficult.[2] For instance, Hausknecht et al. [16] show that a state-of-the-art model achieves a mere $2.56\%$ of the total possible score on a curated set of text-based games for human players [5]. On the other hand, while text-based games exhibit many of the same difficulties as linguistic tasks like open-ended dialogue, they are more structured and constrained.

To design successful agents for text-based games, previous works have relied largely on heuristics that exploit games' inherent structure. For example, several works have proposed rule-based components

---

[*]   Equal contribution.

[2]We challenge readers to solve this representative game: `https://aka.ms/textworld-tryit`.

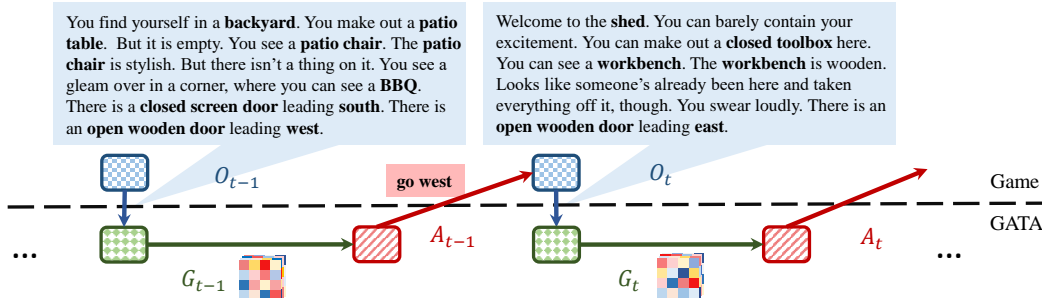

You find yourself in a **backyard**. You make out a **patio table**. But it is empty. You see a **patio chair**. The **patio chair** is stylish. But there isn't a thing on it. You see a gleam over in a corner, where you can see a **BBQ**. There is a **closed screen door** leading **south**. There is an **open wooden door** leading **west**.

Welcome to the **shed**. You can barely contain your excitement. You can make out a **closed toolbox** here. You can see a **workbench**. The **workbench** is wooden. Looks like someone's already been here and taken everything off it, though. You swear loudly. There is an **open wooden door** leading **east**.

Figure 1: GATA playing a text-based game by updating its belief graph. In response to action $A_{t-1}$, the environment returns text observation $O_t$. Based on $O_t$ and $\mathcal{G}_{t-1}$, the agent updates $\mathcal{G}_t$ and selects a new action $A_t$. In the figure, blue box with squares is the game engine, green box with diamonds is the graph updater, red box with slashes is the action selector.

that prune the action space or shape the rewards according to *a priori* knowledge of the game dynamics [50, 24, 1, 48]. More recent approaches take advantage of the graph-like structure of text-based games by building knowledge graph (KG) representations of the game state: Ammanabrolu and Riedl [4], Ammanabrolu and Hausknecht [3], for example, use hand-crafted heuristics to populate a KG that feeds into a deep neural agent to inform its policy. Despite progress along this line, we expect more general, effective representations for text-based games to arise in agents that learn and scale more automatically, which replace heuristics with learning [37].

This work investigates how we can learn graph-structured state representations for text-based games in an entirely data-driven manner. We propose the graph aided transformer agent (GATA)[3] that, in lieu of heuristics, *learns* to construct and update graph-structured beliefs[4] and use them to further optimize rewards. We introduce two self-supervised learning strategies—based on text reconstruction and mutual information maximization—which enable our agent to learn latent graph representations without direct supervision or hand-crafted heuristics.

We benchmark GATA on 500+ unique games generated by TextWorld [9], evaluating performance in a setting that requires generalization across different game configurations. We show that GATA outperforms strong baselines, including text-based models with recurrent policies. In addition, we compare GATA to agents with access to ground-truth graph representations of the game state. We show that GATA achieves competitive performance against these baselines even though it receives only partial text observations of the state. Our findings suggest, promisingly, that graph-structured representations provide a useful inductive bias for learning and generalizing in text-based games, and act as a memory enabling agents to optimize rewards in a partially observed setting.

## 2 Background

**Text-based Games:** Text-based games can be formally described as partially observable Markov decision processes (POMDPs) [9]. They are environments in which the player receives text-only observations $O_t$ (these describe the observable state, typically only partially) and interacts by issuing short text phrases as actions $A_t$ (e.g., in Figure 1, `go west` moves the player to a new location). Often, the end goal is not clear from the start; the agent must infer the objective by earning sparse rewards for completing subgoals. Text-based games have a variety of difficulty levels determined mainly by the environment's complexity (i.e., how many locations in the game, and how many objects are interactive), the game length (i.e., optimally, how many actions are required to win), and the verbosity (i.e., how much text information is irrelevant to solving the game).

**Problem Setting:** We use TextWorld [9] to generate unique *choice-based* games of varying difficulty. All games share the same overarching theme: an agent must gather and process cooking ingredients, placed randomly across multiple locations, according to a recipe it discovers during the game. The agent earns a point for collecting each ingredient and for processing it correctly. The game is won

upon completing the recipe. Processing any ingredient incorrectly terminates the game (e.g., `slice carrot` when the recipe asked for a *diced carrot*). To process ingredients, an agent must find and use appropriate tools (e.g., a knife to `slice`, `dice`, or `chop`; a stove to `fry`, an oven to `roast`).

We divide generated games, all of which have unique recipes and map configurations, into sets for training, validation, and test. Adopting the supervised learning paradigm for evaluating generalization, we tune hyperparameters on the validation set and report performance on a test set of previously unseen games. Testing agents on unseen games (within a difficulty level) is uncommon in prior RL work, where it is standard to train and test on a single game instance. Our approach enables us to measure the robustness of learned policies as they generalize (or fail to) across a "distribution" of related but distinct games. Throughout the paper, we use the term *generalization* to imply the ability of a single policy to play a distribution of related games (within a particular difficulty level).

**Graphs and Text-based Games:** We expect graph-based representations to be effective for text-based games because the state in these games adheres to a graph-like structure. The essential content in most observations of the environment corresponds either to entity attributes (e.g., the state of the `carrot` is `sliced`) or to relational information about entities in the environment (e.g., the `kitchen` is `north_of` the `bedroom`). This information is naturally represented as a dynamic graph $\mathcal{G}_t = (\mathcal{V}_t, \mathcal{E}_t)$, where the vertices $\mathcal{V}_t$ represent entities (including the player, objects, and locations) and their current conditions (e.g., closed, fried, sliced), while the edges $\mathcal{E}_t$ represent relations between entities (e.g., `north_of`, `in`, `is`) that hold at a particular time-step $t$. By design, in fact, the full state of any game generated by TextWorld can be represented explicitly as a graph of this type [53]. The aim of our model, GATA, is to estimate the game state by learning to build graph-structured beliefs from raw text observations. In our experiments, we benchmark GATA against models with direct access to the ground-truth game state rather than GATA's noisy estimate thereof inferred from text.

# 3   Graph Aided Transformer Agent (GATA)

In this section, we introduce GATA, a novel transformer-based neural agent that can infer a graph-structured belief state and use that state to guide action selection in text-based games. As shown in Figure 2, the agent consists of two main modules: a graph updater and an action selector. [5] At game step $t$, the graph updater extracts relevant information from text observation $O_t$ and updates its belief graph $\mathcal{G}_t$ accordingly. The action selector issues action $A_t$ conditioned on $O_t$ and the belief graph $\mathcal{G}_t$. Figure 1 illustrates the interaction between GATA and a text-based game.

## 3.1   Belief Graph

We denote by $\mathcal{G}$ a belief graph representing the agent's belief about the true game state according to what it has observed so far. We instantiate $\mathcal{G} \in [-1, 1]^{\mathcal{R} \times \mathcal{N} \times \mathcal{N}}$ as a real-valued adjacency tensor, where $\mathcal{R}$ and $\mathcal{N}$ indicate the number of relation types and entities. Each entry $\{r, i, j\}$ in $\mathcal{G}$ indicates the strength of an inferred relationship $r$ from entity $i$ to entity $j$. We select $\mathcal{R} = 10$ and $\mathcal{N} = 99$ to match the maximum number of relations and entities in our TextWorld-generated games. In other words, we assume that GATA has access to the *vocabularies* of possible relations and entities but it must learn the structure among these objects, and their semantics, from scratch.

## 3.2   Graph Updater

The graph updater constructs and updates the dynamic belief graph $\mathcal{G}$ from text observations $O_t$. Rather than generating the entire belief graph at each step $t$, we generate a graph update, $\Delta g_t$, that represents the change of the agent's belief after receiving a new observation. This is motivated by the fact that observations $O_t$ typically communicate only incremental information about the state's change from time step $t - 1$ to $t$. The relation between $\Delta g_t$ and $\mathcal{G}$ is given by

$$\mathcal{G}_t = \mathcal{G}_{t-1} \oplus \Delta g_t, \tag{1}$$

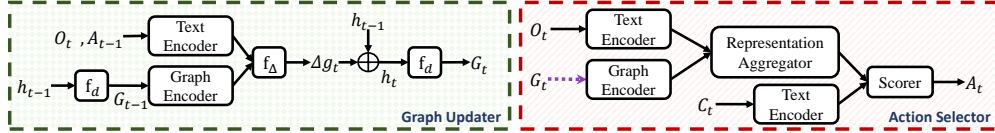

Figure 2: GATA in detail. The coloring scheme is same as in Figure 1. The graph updater first generates $\Delta g_t$ using $\mathcal{G}_{t-1}$ and $O_t$. Afterwards the action selector uses $O_t$ and the updated graph $\mathcal{G}_t$ to select $A_t$ from the list of action candidates $C_t$. Purple dotted line indicates a detached connection (i.e., no back-propagation through such connection).

where $\oplus$ is a graph operation function that produces the new belief graph $\mathcal{G}_t$ given $\mathcal{G}_{t-1}$ and $\Delta g_t$. We formulate the graph operation function $\oplus$ using a recurrent neural network (e.g., a GRU [8]) as:

$$
\begin{aligned}
\Delta g_t &= \mathrm{f}_\Delta(h_{\mathcal{G}_{t-1}}, h_{O_t}, h_{A_{t-1}}); \\
h_t &= \mathrm{RNN}(\Delta g_t, h_{t-1}); \\
\mathcal{G}_t &= \mathrm{f}_\mathrm{d}(h_t).
\end{aligned}
\tag{2}
$$

The function $\mathrm{f}_\Delta$ aggregates the information in $\mathcal{G}_{t-1}$, $A_{t-1}$, and $O_t$ to generate the graph update $\Delta g_t$. $h_{\mathcal{G}_{t-1}}$ denotes the representation of $\mathcal{G}_{t-1}$ from the graph encoder. $h_{O_t}$ and $h_{A_{t-1}}$ are outputs of the text encoder (refer to Figure 2, left part). The vector $h_t$ is a recurrent hidden state from which we decode the adjacency tensor $\mathcal{G}_t$; $h_t$ acts as a memory that carries information across game steps—a crucial function for solving POMDPs [15]. The function $\mathrm{f}_\mathrm{d}$ is a multi-layer perceptron (MLP) that decodes the recurrent state $h_t$ into a real-valued adjacency tensor (i.e., the belief graph $\mathcal{G}_t$). We elaborate on each of the sub-modules in Appendix A.

**Training the Graph Updater:** We pre-train the graph updater using two self-supervised training regimes to learn structured game dynamics. After pre-training, the graph updater is fixed during GATA's interaction with games; at this time it provides belief graphs $\mathcal{G}$ to the action selector. We train the action selector subsequently via RL. Both pre-training tasks share the same goal: to ensure that $\mathcal{G}_t$ encodes sufficient information about the environment state at game step $t$. For training data, we gather a collection of transitions by following walkthroughs in *FTWP* games.[6] To ensure variety in the training data, we also randomly sample trajectories off the optimal path. Next we describe our pre-training approaches for the graph updater.

- **Observation Generation (OG):** Our first approach to pre-train the graph updater involves training a decoder model to reconstruct text observations from the belief graph. Conditioned on the belief graph, $\mathcal{G}_t$, and the action performed at the previous game step, $A_{t-1}$, the observation generation task aims to reconstruct $O_t = \{O_t^1, \ldots, O_t^{L_{O_t}}\}$ token by token, where $L_{O_t}$ is the length of $O_t$. We formulate this task as a sequence-to-sequence (Seq2Seq) problem and use a transformer-based model [43] to generate the output sequence. Specifically, conditioned on $\mathcal{G}_t$ and $A_{t-1}$, the transformer decoder predicts the next token $O_t^i$ given $\{O_t^1, \ldots, O_t^{i-1}\}$. We train the Seq2Seq model using teacher-forcing to optimize the negative log-likelihood loss:

$$
\mathcal{L}_{\mathrm{OG}} = -\sum_{i=1}^{L_{O_t}} \log p_{\mathrm{OG}}(O_t^i | O_t^1, \ldots, O_t^{i-1}, \mathcal{G}_t, A_{t-1}),
\tag{3}
$$

where $p_{\mathrm{OG}}$ is the conditional distribution parametrized by the observation generation model.

- **Contrastive Observation Classification (COC):** Inspired by the literature on contrastive representation learning [41, 19, 44, 7], we reformulate OG mentioned above as a contrastive prediction task. We use contrastive learning to maximize mutual information between the predicted $\mathcal{G}_t$ and the text observations $O_t$. Specifically, we train the model to differentiate between representations corresponding to true observations $O_t$ and "corrupted" observations $\widetilde{O}_t$, conditioned on $\mathcal{G}_t$ and $A_{t-1}$. To obtain corrupted observations, we sample randomly from the set of all collected observations across our pre-training data. We use a noise-contrastive objective and minimize the binary cross-entropy (BCE) loss given by

$$
\mathcal{L}_{\mathrm{COC}} = \frac{1}{K} \sum_{t=1}^{K} \left( \mathbb{E}_O \left[ \log \mathcal{D}\left(h_{O_t}, h_{\mathcal{G}_t}\right) \right] + \mathbb{E}_{\widetilde{O}} \left[ \log\left(1 - \mathcal{D}\left(h_{\widetilde{O}_t}, h_{\mathcal{G}_t}\right)\right) \right] \right).
\tag{4}
$$

Here, $K$ is the length of a trajectory as we sample a positive and negative pair at each step and $\mathcal{D}$ is a *discriminator* that differentiates between positive and negative samples. The motivation behind contrastive unsupervised training is that one does not require to train complex decoders. Specifically, compared to OG, the COC's objective relaxes the need for learning syntactical or grammatical features and allows GATA to focus on learning the semantics of the $O_t$.

We provide further implementation level details on both these self-supervised objectives in Appendix B.

### 3.3 Action Selector

The graph updater discussed in the previous section defines a key component of GATA that enables the model to maintain a structured belief graph based on text observations. The second key component of GATA is the *action selector*, which uses the belief graph $\mathcal{G}_t$ and the text observation $O_t$ at each time-step to select an action. As shown in Figure 2, the action selector consists of four main components: the *text encoder* and *graph encoder* convert text inputs and graph inputs, respectively, into hidden representations; a *representation aggregator* fuses the two representations using an attention mechanism; and a *scorer* ranks all candidate actions based on the aggregated representations.

- **Graph Encoder:** GATA's belief graphs, which estimate the true game state, are multi-relational by design. Therefore, we use relational graph convolutional networks (R-GCNs) [32] to encode the belief graphs from the updater into vector representations. We also adapt the R-GCN model to use embeddings of the available relation labels, so that we can capture semantic correspondences among relations (e.g., `east_of` and `west_of` are reciprocal relations). We do so by learning a vector representation for each relation in the vocabulary that we condition on the word embeddings of the relation's name. We concatenate the resulting vector with the standard node embeddings during R-GCN's message passing phase. Our R-GCN implementation uses basis regularization [32] and highway connections [36] between layers for faster convergence. Details are given in Appendix A.1.

- **Text Encoder:** We adopt a transformer encoder [43] to convert text inputs from $O_t$ and $A_{t-1}$ into contextual vector representations. Details are provided in Appendix A.2.

- **Representation Aggregator:** To combine the text and graph representations, GATA uses a bi-directional attention-based aggregator [49, 33]. Attention from text to graph enables the agent to focus more on nodes that are currently observable, which are generally more relevant; attention from nodes to text enables the agent to focus more on tokens that appear in the graph, which are therefore connected with the player in certain relations. Details are provided in Appendix A.3.

- **Scorer:** The scorer consists of a self-attention layer cascaded with an MLP layer. First, the self-attention layer reinforces the dependency of every token-token pair and node-node pair in the aggregated representations. The resulting vectors are concatenated with the representations of action candidates $C_t$ (from the text encoder), after which the MLP generates a single scalar for every action candidate as a score. Details are provided in Appendix A.4.

**Training the Action Selector:** We use Q-learning [45] to optimize the action selector on reward signals from the training games. Specifically, we use Double DQN [42] combined with multi-step learning [38] and prioritized experience replay [31]. To enable GATA to scale and generalize to multiple games, we adapt standard deep Q-Learning by sampling a new game from the set of training games to collect an episode. Consequently, the replay buffer contains transitions from episodes of different games. We provide further details on this training procedure in Appendix E.2.

### 3.4 Variants Using Ground-Truth Graphs

In GATA, the belief graph is learned entirely from text observations. However, the TextWorld API also provides access to the underlying graph states for games, in the format of discrete KGs. Thus, for comparison, we also consider two models that learn from or encode ground-truth graphs directly.

**GATA-GTP: Pre-training a *discrete* graph updater using ground-truth graphs.** We first consider a model that uses ground-truth graphs to pre-train the graph updater, in lieu of self-supervised methods. GATA-GTP uses ground-truth graphs from *FTWP* during pre-training, but infers belief graphs from the raw text during RL training of the action selector to compare fairly against GATA. Here, the belief graph $\mathcal{G}_t$ is a discrete multi-relational graph. To pre-train a discrete graph updater, we adapt the

command generation approach proposed by Zelinka et al. [53]. We provide details of this approach in Appendix C.

**GATA-GTF: Training the action selector using ground-truth graphs.** To get a sense of the upper bound on performance we might obtain using a belief graph, we also train an agent that uses the full ground-truth graph $\mathcal{G}^{\text{full}}$ during action selection. This agent requires no graph updater module; we simply feed the ground-truth graphs into the action selector (via the graph encoder). The use of ground-truth graphs allows GATA-GTF to escape the error cascades that may result from inferred belief graphs. Note also that the ground-truth graphs contain full state information, relaxing partial observability of the games. Consequently, we expect more effective reward optimization for GATA-GTF compared to other graph-based agents. GATA-GTF's comparison with text-based agents is a sanity check for our hypothesis—that structured representations help learning general policies.

## 4 Experiments and Analysis

We conduct experiments on generated text-based games (Section 2) to answer two key questions:
**Q1:** Does the belief-graph approach aid GATA in achieving high rewards on unseen games after training? In particular, does GATA improve performance compared to SOTA text-based models?
**Q2:** How does GATA compare to models that have access to ground-truth graph representations?

### 4.1 Experimental Setup and Baselines

We divide the games into four subsets with one difficulty level per subset. Each subset contains 100 training, 20 validation, and 20 test games, which are sampled from a distribution determined by their difficulty level. To elaborate on the diversity of games: for easier games, the recipe might only require a single ingredient and the world is limited to a single location, whereas harder games might require an agent to navigate a map of 6 locations to collect and appropriately process up to three ingredients. We also test GATA's transferability across difficulty levels by mixing the four difficulty levels to build level 5. We sample 25 games from each of the four difficulty levels to build a training set. We use all validation and test games from levels 1 to 4 for level 5 validation and test. In all experiments, we select the top-performing agent on validation sets and report its test scores; all validation and test games are unseen in the training set. Statistics of the games are shown in Table 1.

Table 1: Games statistics (averaged across all games within a difficulty level).

| Level | Recipe Size | #Locations | Max Score | Need Cut | Need Cook | #Action Candidates | #Objects |
|-------|-------------|------------|-----------|----------|-----------|--------------------|----------|
| 1 | 1 | 1 | 4 | ✓ | ✗ | 8.9 | 17.1 |
| 2 | 1 | 1 | 5 | ✓ | ✓ | 8.9 | 17.5 |
| 3 | 1 | 9 | 3 | ✗ | ✗ | 4.9 | 34.1 |
| 4 | 3 | 6 | 11 | ✓ | ✓ | 10.8 | 33.4 |
| 5 | | | | Mixture of levels {1,2,3,4} | | | |

As baselines, we use our implementation of LSTM-DQN [29] and LSTM-DRQN [50], both of which use only $O_t$ as input. Note that LSTM-DRQN uses an RNN to enable an implicit memory (i.e., belief); it also uses an episodic counting bonus to encourage exploration [50]. This draws an interesting comparison with GATA, wherein the belief is extracted and updated dynamically, in the form of a graph. For fair comparison, we replace the LSTM-based text encoders with a transformer-based text encoder as in GATA. We denote those agents as Tr-DQN and Tr-DRQN respectively. We denote a Tr-DRQN equipped with the episodic counting bonus as Tr-DRQN+. These three text-based baselines are representative of the current top-performing neural agents on text-based games.

Additionally, we test the variants of GATA that have access to ground-truth graphs (as described in Section 3.4). Comparing with GATA, the GATA-GTP agent also maintains its belief graphs throughout the game; however, its graph updater is pre-trained on *FTWP* using ground-truth graphs—a stronger supervision signal. GATA-GTF, on the other hand, does not have a graph updater. It directly uses ground-truth graphs as input during game playing.

Table 2: Agents' normalized **test** scores and averaged relative improvement ($\% \uparrow$) over Tr-DQN across difficulty levels. An agent m's relative improvement over Tr-DQN is defined as $(R_m - R_{\text{Tr-DQN}})/R_{\text{Tr-DQN}}$ where R is the score. All numbers are percentages. ◆represents ground-truth full graph; ♣represents discrete $\mathcal{G}_t$ generated by GATA-GTP; ♠represents $O_t$. ⋆and ∞are continuous $G_t$ generated by GATA, when the graph updater is pre-trained with OG and COC tasks, respectively.

| Difficulty Level | 20 Training Games | | | | | | 100 Training Games | | | | | | Avg. |
|---|---|---|---|---|---|---|---|---|---|---|---|---|---|
| | 1 | 2 | 3 | 4 | 5 | % ↑ | 1 | 2 | 3 | 4 | 5 | % ↑ | % ↑ |
| Agent | Text-based Baselines | | | | | | | | | | | | |
| Tr-DQN | 66.2 | 26.0 | 16.7 | 18.2 | **27.9** | —— | 62.5 | 32.0 | 38.3 | 17.7 | 34.6 | —— | —— |
| Tr-DRQN | 62.5 | 32.0 | 28.3 | 12.7 | 26.5 | +10.3 | 58.8 | 31.0 | 36.7 | 21.4 | 27.4 | -2.6 | +3.9 |
| Tr-DRQN+ | 65.0 | 30.0 | 35.0 | 11.8 | 18.3 | +10.7 | 58.8 | 33.0 | 33.3 | 19.5 | 30.6 | -3.4 | +3.6 |
| Input | GATA | | | | | | | | | | | | |
| ⋆ | 70.0 | 20.0 | 20.0 | 18.6 | 26.3 | -0.2 | 62.5 | 32.0 | 46.7 | **27.7** | 35.4 | **+16.1** | +8.0 |
| ⋆♠ | 66.2 | **48.0** | 26.7 | 15.5 | 26.3 | +24.8 | **66.2** | **36.0** | 58.3 | 14.1 | 45.0 | **+16.1** | +20.4 |
| ∞ | **73.8** | 42.0 | 26.7 | **20.9** | 24.5 | +27.1 | 62.5 | 30.0 | 51.7 | 23.6 | 36.0 | +13.2 | +20.2 |
| ∞♠ | 68.8 | 33.0 | **41.7** | 17.7 | 27.0 | **+34.9** | 62.5 | 33.0 | 46.7 | 25.9 | 33.4 | +13.6 | **+24.2** |
| | GATA-GTP | | | | | | | | | | | | |
| ♣ | 56.2 | 26.0 | 40.0 | 17.3 | 17.7 | +16.6 | 37.5 | 31.0 | 45.0 | 13.6 | 18.7 | -18.9 | -1.2 |
| ♣♠ | 65.0 | 32.0 | 41.7 | 12.3 | 23.5 | +24.6 | 62.5 | 32.0 | 51.7 | 21.8 | 23.5 | +5.2 | +14.9 |
| | GATA-GTF | | | | | | | | | | | | |
| ◆ | 48.7 | 61.0 | 46.7 | 23.6 | 28.9 | +64.2 | 95.0 | 95.0 | 70.0 | 37.3 | 52.8 | +99.0 | +81.6 |

## Q1: Performance of GATA compared to text-based baselines

In Table 2, we show the normalized test scores achieved by agents trained on either 20 or 100 games for each difficulty level. Equipped with belief graphs, GATA significantly outperforms all text-based baselines. The graph updater pre-trained on both of the self-supervised tasks (Section 3.2) leads to better performance than the baselines (⋆ and ∞). We observe further improvements in GATA's policies when the text observations (♠) are also available. We believe the text observations guide GATA's action scorer to focus on currently observable objects through the bi-attention mechanism. The attention may further help GATA to counteract accumulated errors from the belief graphs. In addition, we observe that Tr-DRQN and Tr-DRQN+ outperform Tr-DQN, with 3.9% and 3.6% relative improvement ($\% \uparrow$). This suggests the implicit memory of the recurrent components improves performance. We also observe GATA substantially outperforms Tr-DQN when trained on 100 games, whereas the DRQN agents struggle to optimize rewards on the larger training sets.

## Q2: Performance of GATA compared to models with access to the ground-truth graph

Table 2 also reports test performance for GATA-GTP (♣) and GATA-GTF (◆). Consistent with GATA, we find GATA-GTP also performs better when given text observations (♠) as additional input to the action scorer. Although GATA-GTP outperforms Tr-DQN by 14.9% when text observations are available, its overall performance is still substantially poorer than GATA. Although the graph updater in GATA-GTP is trained with ground-truth graphs, we believe the discrete belief graphs and the discrete operations for updating them (Appendix C.1) make this approach vulnerable to an accumulation of errors over game steps, as well as errors introduced by the discrete nature of the predictions (e.g., round-off error). In contrast, we suspect that the continuous belief graph and the learned graph operation function (Eqn. 2) are easier to train and recover more gracefully from errors.

Meanwhile, GATA-GTF, which uses ground-truth graphs $\mathcal{G}^{\text{full}}$ during training and testing, obtains significantly higher scores than does GATA and all other baselines. Because $\mathcal{G}^{\text{full}}$ turns the game environment into a fully observable MDP and encodes accurate state information with no error accumulation, GATA-GTF represents the performance upper-bound of all the $\mathcal{G}_t$-based baselines. The scores achieved by GATA-GTF reinforce our intuition that belief graphs improve text-based game

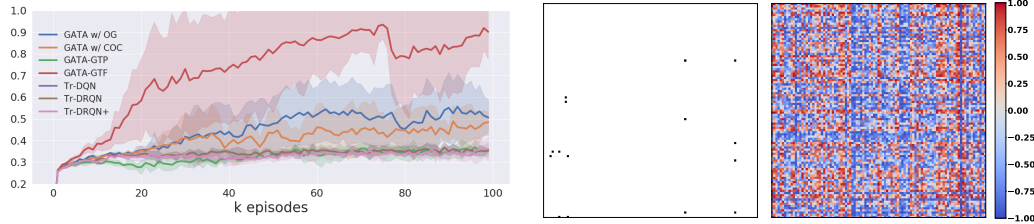

Figure 3: **Left:** Training curves on 20 level 2 games (averaged over 3 seeds). **Right:** Density comparison between a ground-truth graph (binary) and a belief graph $\mathcal{G}$ generated by the COC pre-training procedure. Both matrices are slices of adjacency tensors corresponding the `is` relation.

agents. At the same time, the performance gap between GATA and GATA-GTF invites investigation into better ways to learn accurate graph representations of text.

**Additional Results**

We also show the agents' training curves and examples of the belief graphs $\mathcal{G}$ generated by GATA. Figure 3 (**Left**) shows an example of all agents' training curves. We observe consistent trends with the testing results of Table 2 — GATA outperforms the text-based baselines and GATA-GTP, but a significant gap exists between GATA and GATA-GTF (which uses ground-truth graphs as input to the action scorer). Figure 3 (**Right**) highlights the sparsity of a ground-truth graph compared to that of a belief graph $\mathcal{G}$. Since generation of $\mathcal{G}$ is unsupervised by any ground-truth graphs, we do not expect $\mathcal{G}$ to be interpretable nor sparse. Further, since the self-supervised models learn belief graphs directly from text, some of the learned features may correspond to the underlying grammar or other features useful for the self-supervised tasks, rather than only being indicative of relationships between objects. However, we show $\mathcal{G}$ encodes useful information for a relation prediction probing task in Appendix D.5.

Given space limitations, we only report a representative selection of our results in this section. Appendix D provides an exhaustive set of results including training curves, training scores, and test scores for all experimental settings introduced in this work. We also provide a detailed qualitative analysis including hi-res visualizations of the belief graphs. We encourage readers to refer to it.

## 5  Related Work

**Dynamic graph extraction:**  Numerous recent works have focused on constructing graphs to encode structured representations of raw data, for various tasks. Kipf et al. [23] propose contrastive methods to learn latent structured world models (C-SWMs) as state representations for vision-based environments. Their work, however, does not focus on learning policies to play games or to generalize across varying environments. Das et al. [10] leverage a machine reading comprehension mechanism to query for entities and states in short text passages and use a dynamic graph structure to track changing entity states. Fan et al. [12] propose to encode graph representations by linearizing the graph as an input sequence in NLP tasks. Johnson [21] construct graphs from text data using gated graph transformer neural networks. Yang et al. [46] learn transferable latent relational graphs from raw data in a self-supervised manner. Compared to the existing literature, our work aims to infer multi-relational KGs dynamically from partial text observations of the state and subsequently use these graphs to inform general policies. Concurrently, Srinivas et al. [35] propose to learn state representations with contrastive learning methods to facilitate RL training. However, they focus on vision-based environments and they do not investigate generalization.

More generally, we want to note that compared to traditional knowledge base construction (KBC) works, our approach is more related to the direction of neural relational inference [22]. In particular, we seek to generate task-specific graphs, which tend to be dynamic, contextual and relatively small, whereas traditional KBC focus on generating large, static graphs.

**Playing Text-based Games:**  Recent years have seen a host of work on playing text-based games. Various deep learning agents have been explored [29, 17, 14, 51, 20, 3, 52, 47]. Fulda et al. [13] use pre-trained embeddings to reduce the action space. Zahavy et al. [51], Seurin et al. [34], and

Jain et al. [20] explicitly condition an agent's decisions on game feedback. Most of this literature trains and tests on a single game without considering generalization. Urbanek et al. [40] use memory networks and ranking systems to tackle adventure-themed dialog tasks. Yuan et al. [50] propose a count-based memory to explore and generalize on simple unseen text-based games. Madotto et al. [26] use GoExplore [11] with imitation learning to generalize. Adolphs and Hofmann [1] and Yin and May [48] also investigate the multi-game setting. These methods rely either on reward shaping by heuristics, imitation learning, or rule-based features as inputs. We aim to minimize hand-crafting, so our action selector is optimized only using raw rewards from games while other components of our model are pre-trained on related data. Recently, Ammanabrolu and Riedl [4], Ammanabrolu and Hausknecht [3], Yin and May [48] leverage graph structure by using rule-based, untrained mechanisms to construct KGs to play text-based games.

## 6 Conclusion

In this work, we investigate how an RL agent can play and generalize within a distribution of text-based games using graph-structured representations inferred from text. We introduce GATA, a novel neural agent that infers and updates latent belief graphs as it plays text-based games. We use a combination of RL and self-supervised learning to teach the agent to encode essential dynamics of the environment in its belief graphs. We show that GATA achieves good test performance, outperforming a set of strong baselines including agents pre-trained with ground-truth graphs. This evinces the effectiveness of generating graph-structured representations for text-based games.

## 7 Broader Impact

Our work's immediate aim—improved performance on text-based games—might have limited consequences for society; however, taking a broader view of our work and where we'd like to take it forces us to consider several social and ethical concerns. We use text-based games as a proxy to model and study the interaction of machines with the human world, through language. Any system that interacts with the human world impacts it. As mentioned previously, an example of language-mediated, human-machine interaction is online customer service systems.

- In these systems, especially in products related to critical needs like healthcare, providing inaccurate information could result in serious harm to users. Likewise, failing to communicate clearly, sensibly, or convincingly might also cause harm. It could waste users' precious time and diminish their trust.

- The responses generated by such systems must be inclusive and free of bias. They must not cause harm by the act of communication itself, nor by making decisions that disenfranchise certain user groups. Unfortunately, many data-driven, free-form language generation systems currently exhibit bias and/or produce problematic outputs.

- Users' privacy is also a concern in this setting. Mechanisms must be put in place to protect it. Agents that interact with humans almost invariably train on human data; their function requires that they solicit, store, and act upon sensitive user information (especially in the healthcare scenario envisioned above). Therefore, privacy protections must be implemented throughout the agent development cycle, including data collection, training, and deployment.

- Tasks that require human interaction through language are currently performed by people. As a result, advances in language-based agents may eventually displace or disrupt human jobs. This is a clear negative impact.

Even more broadly, any systems that generate convincing natural language could be used to spread misinformation.

Our work is immediately aimed at improving the performance of RL agents in text-based games, in which agents must understand and act in the world through language. Our hope is that this work, by introducing graph-structured representations, endows language-based agents with greater accuracy and clarity, and the ability to make better decisions. Similarly, we expect that graph-structured representations could be used to constrain agent decisions and outputs, for improved safety. Finally, we believe that structured representations can improve neural agents' interpretability to researchers and users. This is an important future direction that can contribute to accountability and transparency

in AI. As we have outlined, however, this and future work must be undertaken with awareness of its hazards.

# 8  Acknowledgements

We thank Alessandro Sordoni and Devon Hjelm for the helpful discussions about the probing task. We also thank David Krueger, Devendra Singh Sachan, Harm van Seijen, Harshita Sahijwani, Jacob Miller, Koustuv Sinha, Loren Lugosch, Meng Qu, Travis LaCroix, and the anonymous ICML 2020 and NeurIPS 2020 reviewers and ACs for their insightful comments on an earlier draft of this work. The work was funded in part by an academic grant from Microsoft Research, an NSERC Discovery Grant RGPIN-2019-05123, an IVADO Fundamental Research Project Grant PRF-2019-3583139727, as well as Canada CIFAR Chairs in AI, held by Prof. Hamilton, Prof. Poupart and Prof. Tang.

## Footnotes

[3]Code and dataset used: `https://github.com/xingdi-eric-yuan/GATA-public`

[4]Text-based games are partially observable environments.

[5]The graph updater and action selector share some structures but not their parameters (unless specified).

[6]This is an independent and unique set of TextWorld games [39]. Details are provided in Appendix F.

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
