[Reviews · NeurIPS 2020]

Review 1

Summary and Contributions: The work investigates inferring graph structured state representations from text-game observations and learning an action selector network to subsequently learning to play said games effectively. The graphs are defined over a number of entities N and relations R (dimensionality = R x N x N). The authors introduce an agent that can compose belief graphs from text observations and acting over them: Graph Aided Transformer Agent (GATA). This model generates new graphs given new text observations via step-wise graph updates The graphs are pre-trained in two phases: Observation Generation: Reconstruct observations token-by-token from the belief graph and build a loss from that. Contrastive Observation Classification: Contrastive Loss by using G_t to predict between the true token and a corrupted version over trajectories of length K. In this way a belief graph is generated. Next, an action selector uses the pre-trained belief graph and the text observations per timestep to take action. The action selector consists of four components. Encoders for both graphs and text observations to generate encoded representations via graph convolutional networks and a transformer network respectively. The representations are combined via an aggregator using attention and then the actions are scored via an MLP. The entire stack is trained via Deep Q-Learning. GATA was evaluated on the TextWorld environment with over 500 unique games where the player is required to assemble ingredients from various locations and make a recipe. The authors investigate 4 difficulty levels of the game increasing with bigger recipes and locations up to five levels. The authors set out to answer two questions 1) Does GATA perform well on unseen games and improve SOTA? 2) How does GATA compare to models that have access to ground truth models of the graph. The authors demonstrate that their method outperforms existing approaches and that when augmented with ground truth graphs they can achieve even greater performance.

Strengths: In the experimental results the performance of GATA over the baselines is significant and broad and demonstrates this models advantage over the baselines clearly. This work also investigates the effect of extracting pre-training discrete graphs and training the action selector given ground truth graphs which I believe is key to contextualizing the power of this approach and demonstrating its soundness. The experiments and approach are motivated by two well formed questions regarding how well GATA generalizes across unseen games and performs in comparison to the state of the art and also the effect of training with ground truth graphs. This gave clarity to the empirical motivation and authors went about answering these questions clearly as well.

Weaknesses: Each of the baselines are LSTM-DQN variants and it would have been interesting to see comparisons to other baselines if possible. In particular, in the introduction it is mentioned that many previous approaches have relied on heuristics given a priori knowledge. Seeing GATA's performance alongside these approaches in an appropriate domain would be interesting to see and I believe strengthen the case for this work. It would have been interesting to see more along the lines of analysis of the knowledge encoded in the graphs. Fig. 3 includes some of this but it's not entirely clear what the visualization represents (ie. what do they entities and relations maybe mean?). A bit more breadth in the tested environments might have been interesting to see. Possibly evaluating on different types of text games (I'm less familiar with what might be standard here) or across another NLP task that fits well to the domain. Or even more levels of difficulty. That said, I think the domain chosen sufficiently demonstrates the value of this work.

Correctness: The authors clearly laid out the methodology of their work, described a novel agent and a pre-training scheme that they used to do so and presented convincing results. Some comparison to other baselinesmay have strengthened their claim.

Clarity: The paper is well written.

Relation to Prior Work: The authors use dynamic graph extraction applied in a novel way to build an agent well suited to solving text based games generally. This is a combines approaches from graph extraction, transformer models and approaches to solving text based games. Notably, this is done without heuristics or any hand-crafted features.

Reproducibility: Yes

Additional Feedback:


Review 2

Summary and Contributions: This paper presents GATA, a graph transformer agents that plays text-adventure games. Specifically, GATA learns how to construct the graph representation of the world state instead of using hand-coded heuristics as in previous state-of-the-art text-adventure playing agents. The agent is tested in TextWorld, where it must play variations of the cooking game that are previously unseen.

Strengths: Soundness: The technique of learning to construct the graph requires pre-training, which means relevant pre-training data must be available. However, if available, the technique looks sound and is theoretically motivated. Prior graph-based text-adventure playing agents have heuristic rules for extracting graph structure from text observations. This system avoids that bottleneck at the expense of pretraining. Evaluation: the authors should be commended for training and testing on separate game world maps. The authors refer to this as generalization, although one could also call it transfer (albeit near-transfer). This should be how all RL agents are evaluated. Relevance to NeurIPS: the work is significant in the extent that text-adventure games are increasingly seen as a stepping stone for agents that might need to one day use language to act in the real world (such as dialogue). Significance: The work overcomes the bottleneck of having to use heuristic graph update rules. However, one limitation of the significance is that the TextWorld domain is rather simple, and many other researchers working on text-game playing agents are using Zork and other full, commercial games in the Jericho package. Evaluation: The strengths of the evaluation is, as noted, testing under near-transfer conditions. The agent is also tested against two versions that have more complete knowledge. These upper-bounds are rarely presented (although one of the "upper bounds" ended up not being an upper bound). This gives a good sense for not just how much progress has been made, but how much more there is to go. This can give new researchers some idea of the direction for new research attempts.

Weaknesses: Evaluation: the evaluation methodology has some downsides. In particular, the GATA agent is not evaluated against KG-DQN or KG-A2C which are the state-of-the-art agents using graph based representations of world state. Granted they are using heuristics to build their graphs so the comparison is not apples to apples. However, there is a missed opportunity to look at the "perfect-information" versions of GATA, ie GATA-GTF and GATA-GTP against KG-A2C or KG-DQN. Or even construct KG-DQN-GTF to give perfectly even playing field.

Correctness: The technique appears correct. The empirical methodology is sufficient though there are some weaknesses noted above.

Clarity: The paper is clearly written. The motivation for Contrastive Observation Classification is unclear. What problem does this solve, or what would happen if this step is not done?

Relation to Prior Work: It is sufficient.

Reproducibility: Yes

Additional Feedback: What is described as generalization may also be described as transfer, albeit a relatively near-transfer. However, since RL agents notoriously overfit to their environments, even near-transfer can be challenging. The fact that the internal representation of text-games are graph-like is not necessarily the reason why graph-based neural networks should work well. Playing the game requires memory of states because of the particular nature of room-based partial observability. Knowledge graphs are structured memory.


Review 3

Summary and Contributions: The paper presents a transformer-based model that learns to construct a belief graph to facilitate policy learning in text-based games. A graph updater, implemented as an end-to-end neural network, is pre-trained to represent the relation-entity-entity tuples from textual observations, where relations and entities are given as in a closed-domain setting. Then an action selector, implemented as a neural network, aggregates the graph information and the textual information to learn a value function for policy learning. The proposed method outperformed text-only methods on the TextWorld domains.

Strengths: The paper investigates learning approaches to automatically construct and update knowledge graphs to address the partial observability for text-based games. The partial observability issue is general and common for text-based sequential decision making problems. The proposed technique may be utilized by various domains. The paper has an extensive empirical section (and supplemental materials). Empirically, the proposed method outperforms competitive baselines that consider textual observations. Sufficient ablative study results are provided as well.

Weaknesses: The knowledge graph representation is in the form of three-way tensors. This tensor-based representation may be hard to scale for practical text-based domains. The situation may be worsened when they are required to be loaded into GPUs. The paper's approach freezes the graph updater after pre-training, so the graph building part can be viewed as an information extraction module to convert the textual information into a knowledge graph. It would be good if some discussion can be provided on the advantage of the proposed approach compared to the standard NLP information extraction methods, such as OpenIE 5[1] or some deep learning based one [2]. It seems that the required knowledge graph representation can be incrementally constructed via these information extraction methods with some filters. [1] Schmitz, Michael, et al. "Open language learning for information extraction." Proceedings of the 2012 Joint Conference on Empirical Methods in Natural Language Processing and Computational Natural Language Learning. 2012. [2] Stanovsky, Gabriel, et al. "Supervised open information extraction." Proceedings of the 2018 Conference of the North American Chapter of the Association for Computational Linguistics: Human Language Technologies, Volume 1 (Long Papers). 2018. ** ** The author's responses are very useful and they clarified the scalability and the task-specific data-driven information extraction contribution of the proposed method. I have increased my score accordingly.

Correctness: YES

Clarity: YES

Relation to Prior Work: [1] Schmitz, Michael, et al. "Open language learning for information extraction." Proceedings of the 2012 Joint Conference on Empirical Methods in Natural Language Processing and Computational Natural Language Learning. 2012. [2] Stanovsky, Gabriel, et al. "Supervised open information extraction." Proceedings of the 2018 Conference of the North American Chapter of the Association for Computational Linguistics: Human Language Technologies, Volume 1 (Long Papers). 2018.

Reproducibility: Yes

Additional Feedback:


Review 4

Summary and Contributions: This paper proposes a transformer-based model that updates a belief network in order to aid in playing text-based adventure games. This method is evaluated on games with varying degrees of difficulty and, more importantly, on games that the agent wasn't initially trained on. This further serves showcasing how well the model can generalize to new, adjacent tasks.

Strengths: The methods were well motivated and justified for the given problem. The experiments were well laid out and had plenty of ablations done within them to further bolster their findings. Incorporating a belief network for the agent to dynamically update and reference is definitely appropriate for the task and novel.

Weaknesses: I identified no weaknesses along those axes mentioned.

Correctness: The methodology appeared to be correct to me.

Clarity: The paper was well written. As someone who isn't the most up to date on reinforcement learning topics I was able to follow along just fine and take away the major ideas without any issues. The only one issue I had was in reading the findings of Table 2. The symbols used were a little confusing. I understand it was to save space, but it took a lot of staring in order to match up the meanings of everything. Maybe the table could use a bit of iterating in order to make it more digestible upon first glance?

Relation to Prior Work: Yes, this work is clearly positioned in relation to prior work.

Reproducibility: Yes

Additional Feedback:

[Author Response · NeurIPS 2020]

We thank all the reviewers for their insightful comments, suggestions, and positive feedback. We will update the paper
based on the constructive comments provided. We start by addressing a common point regarding the use of OpenIE
methods for text-based games, and then address questions from individual reviewers.

**Comparison with OpenIE-based agents.** One common point raised in the reviews is the relationship between our
approach and approaches that use OpenIE algorithms for graph construction, such as KG-DQN [4] and KG-A2C [3].
These are indeed important prior works, and the key distinction between GATA (our work) and [4] and [3] is that they
rely on handcrafted filters combined with OpenIE algorithms, whereas GATA is entirely data-driven. We actually put
significant effort into including [4] as a baseline in the months prior to our submission. However, unfortunately, we
were unable to faithfully reproduce their model or achieve non-trivial performance on our evaluation games. These
problems persisted even after contacting the original authors for assistance, and we believe two key issues were (a)
the hand-crafted filters in [4] not generalizing across games and (b) substantial scalability issues. More generally,
off-the-shelf OpenIE tools are suboptimal for maintaining dynamic knowledge graphs in text-based games; in particular:
• Low precision: OpenIE generates a large amount of unrelated tuples (e.g., from a sentence "what is that?" it extracts
nodes "what" and "that" connected by edge "is"). Empirically, a set of filtering heuristics are needed as used in [3, 4].
• Low recall: Tools like OpenIE can hardly cover certain node types appearing in text-based games. Some nodes that
describe states of objects are absent from $O_t$. For instance, the resulting observation of making a meal will be "you
made a meal", but the "consumed" states of the ingredients are absent from $O_t$ thus require reasoning. Typically, such
types of nodes are not entities (nor even noun phrases), making it difficult for tools like OpenIE to detect them.
• Dynamicity: Since we are dealing with a dynamic environment, some handcrafted heuristics for maintaining/updating
the knowledge graph (tuples extracted by OpenIE) are needed. For instance, after moving to a new room, the content of
the KG shouldn't be discarded but some relations need to be changed (e.g., location of the player).

**Reviewer #1:**
Analysing generated graphs: R1 raises an important point on analysing the graphs generated. For the analyses of
information encoded by the graphs, we refer them to Appendix D.5. Here, apart from the high-res visualization of the
graphs, we conduct probing tasks to study the information encoded by the generated belief graphs. Results suggest the
graphs can encode important information (Table 7). We will further discuss these results in the revision.
Other games: An alternative testbed would be to use the curated list of text-based games supported by Jericho [14].
However, the list is small with only 50 highly diverse games. Attempting to solve the very difficult problem of
out-of-distribution generalization (OODG) on diverse Jericho games was out-of-scope for this current submission,
which focuses on achieving state-of-the-art results on in-domain generalization. However, improving on GATA to
tackle Jericho and other more challenging generalization tasks is an objective of future work.

**Reviewer #2:**
Motivation for COC: Both the graph updater pre-training approaches (OG and COC) aim to extract useful information
that is sufficient to reconstruct the text observation. Contrastive learning has been shown to be an effective unsupervised
training regime to learn useful information in several prior works [7, 43, 22]. The motivation behind contrastive
unsupervised training is that one does not require to train complex decoders. For example, compared to OG, the COC's
objective relaxes the need for learning syntactical or grammatical features and allows GATA to focus on learning the
semantics of the $O_t$. This is an important point, and we will clarify this motivation in the revised main text.
Graph as memory: We agree with the reviewer that a graph-based representation also serves as a structured memory for
the agent. In particular, we believe such structured representations (KG) generalize well for text-based games owing to
two reasons: a) they serve as a good inductive bias as the states can be factorized in terms of nodes and relations; and b)
they act as a memory which aids the agent to act effectively in a partially observed scenario.

**Reviewer #3:**
Scalability: You raise some important clarification points regarding scalability. One important clarification is that at
every step, the belief graph $\mathcal{G} \in [-1, 1]^{\mathcal{R} \times \mathcal{N} \times \mathcal{N}}$ is decoded using $f_d$ from a single vector $h_t$ (as illustrated in Fig 2), and
the size of this vector $h_t$ is significantly smaller than the full tensor (i.e., dimension 64, as described in appendices). We
use this low-dimensional vector as the model's recurrent state, which alleviates scalability issues during learning. More
generally, regarding scalability and relationships to traditional knowledge base construction (KBC), it is important to
note that our approach is more related to recent work on neural relational inference (NRI, Kipf et al., arXiv:1802.04687)
than traditional KBC; in particular, we seek to generate task-specific graphs, which tend to be dynamic, contextual and
relatively small, whereas traditional KBC focus on generating large, static graphs. As a result, we believe discussing
extrapolations of GATA's graph extraction to IE domains is out of scope of this paper.
OpenIE: Please see our general response. Thank you for pointing this out and providing the references, we acknowledge
that our approach is related to information extraction literature and we will discuss them in our revision.

**Reviewer #4:**
Table 2: We will make the presentation of our experiments more clear in the revised version.

[Meta-Review · NeurIPS 2020]

The paper presents a transformer-based agent that can learn to play text adventure games. The reviewers are generally positive towards the paper. The method is well evaluated through ablation studies, and no significant shortcomings were found.